# Lateral Flow Immunoassay with Quantum-Dot-Embedded Silica Nanoparticles for Prostate-Specific Antigen Detection

**DOI:** 10.3390/nano12010033

**Published:** 2021-12-23

**Authors:** Sungje Bock, Hyung-Mo Kim, Jaehi Kim, Jaehyun An, Yun-Sik Choi, Xuan-Hung Pham, Ahla Jo, Kyeong-min Ham, Hobeom Song, Jung-Won Kim, Eunil Hahm, Won-Yeop Rho, Sang Hun Lee, Seung-min Park, Sangchul Lee, Dae Hong Jeong, Ho-Young Lee, Bong-Hyun Jun

**Affiliations:** 1Department of Bioscience and Biotechnology, Konkuk University, Seoul 05029, Korea; bsj4126@konkuk.ac.kr (S.B.); hmkim0109@konkuk.ac.kr (H.-M.K.); susia45@gmail.com (J.K.); ghj4067@konkuk.ac.kr (J.A.); phamricky@gmail.com (X.-H.P.); iamara0421@konkuk.ac.kr (A.J.); hkm7321@konkuk.ac.kr (K.-m.H.); greenice@konkuk.ac.kr (E.H.); 2BioSquare Inc., Hwaseong 18449, Korea; hbsong@bio-square.com (H.S.); jwkim@bio-square.com (J.-W.K.); 3Department of Chemistry Education, Seoul National University, Seoul 08826, Korea; 71388c@naver.com (Y.-S.C.); jeongdh@snu.ac.kr (D.H.J.); 4School of International Engineering and Science, Jeonbuk National University, Jeonju 54896, Korea; rho7272@jbnu.ac.kr; 5Department of Chemical and Biological Engineering, Hanbat National University, Deajeon 34158, Korea; sanghunlee@hanbat.ac.kr; 6Department of Urology, Stanford University School of Medicine, Palo Alto, CA 94305, USA; sp293@stanford.edu; 7Department of Urology, Seoul National University Bundang Hospital, Seongnam 13620, Korea; slee@snubh.org; 8Department of Nuclear Medicine, Seoul National University Bundang Hospital, Seongnam 13620, Korea

**Keywords:** prostate-specific antigen, prostate cancer, lateral flow immunoassay, quantum dot, quantum-dot-embedded silica nanoparticles

## Abstract

Prostate cancer can be detected early by testing the presence of prostate-specific antigen (PSA) in the blood. Lateral flow immunoassay (LFIA) has been used because it is cost effective and easy to use and also has a rapid sample-to-answer process. Quantum dots (QDs) with very bright fluorescence have been previously used to improve the detection sensitivity of LFIAs. In the current study, a highly sensitive LFIA kit was devised using QD-embedded silica nanoparticles. In the present study, only a smartphone and a computer software program, ImageJ, were used, because the developed system had high sensitivity by using very bright nanoprobes. The limit of PSA detection of the developed LFIA system was 0.138 ng/mL. The area under the curve of this system was calculated as 0.852. The system did not show any false-negative result when 47 human serum samples were analyzed; it only detected PSA and did not detect alpha-fetoprotein and newborn calf serum in the samples. Additionally, fluorescence was maintained on the strip for 10 d after the test. With its high sensitivity and convenience, the devised LFIA kit can be used for the diagnosis of prostate cancer.

## 1. Introduction

Globally, cancer is still one of the most life-threatening diseases, with high incidence and mortality rates [1]. Among the numerous types of cancers, prostate cancer has the highest incidence rate (26%) and second-highest mortality rate (22%) among men worldwide [1]. Therefore, diagnosing prostate cancer is crucial for ensuring proper healthcare for men. Many studies on the diagnosis of prostate cancer have been conducted [2,3]. Early diagnosis of prostate cancer is considered essential because it can lower the mortality rate through early treatment [4]. For the early diagnosis of prostate cancer, the expression of prostate-specific antigen (PSA), which is a protein released from the prostate of the cancer patient, has been used as a biomarker to predict the progression of this cancer [5,6]. Generally, a level <2.5 ng/mL PSA is considered normal (safe group), 2.6–4.0 ng/mL PSA requires consultation with a doctor (safe for the most group), 4–10 ng/mL PSA indicates a 25% chance of having prostate cancer (suspect group), and patients with >10 ng/mL PSA in their blood must undergo further clinical tests because they have a 50% probability of having this disease [7]. When the level of PSA in the blood is 2.5 ng/mL or less, the patient is considered to belong to the safe group; thus, it is important to measure the PSA level with a threshold value of 2.5 ng/mL through a simple test with high accuracy because additional tests are not required for this group. If the PSA level is less than 2.5 ng/mL, screening every two years is sufficient [8].

To analyze PSA levels in the blood for the diagnosis of prostate cancer, the enzyme-linked immunosorbent assay (ELISA) method is widely used [9,10,11]. Protein detection through ELISA has many advantages, such as a low limit of detection (LOD; ~10 pg/mL) [12], quantifiability, and high accuracy, selectivity, and reproducibility [13]. The disadvantages of ELISA include a long reaction duration (4–6 h) and the need for a laboratory with trained technicians [14]. To overcome the shortcomings of ELISA, lateral flow immunoassay (LFIA) based on a test strip has been recently proposed as an alternative [15,16,17]. LFIA is used as a point-of-care test with the advantages of fast turnaround time, low cost, and feasibility [18,19,20]. Various types of nanomaterials have been used as detection probes for LFIAs depending on the application [21,22,23]. Metal nanoparticles (NPs), such as those composed of gold (Au), silver (Ag), and alloy metals, have been used for colorimetric detection with the naked eye [21,22,24,25]. Well-designed fluorescent NPs have been used for more sensitive and accurate analysis than colorimetric detection [26,27]. Quantum dots (QDs) are markedly brighter than other fluorescent NPs and have no photobleaching characteristics [28]; thus, QDs have been used for fabricating sensitive LFIA systems [23,29]. Nanostructures with a large number of QDs have stronger fluorescence intensity than those using a single QD. Li et al. encapsulated QDs to prepare QD nanobeads and applied them to LFIA systems for PSA detection (LOD = 0.33 ng/mL) [30]. However, additional equipment is usually required for quantitative analysis in most LFIAs [30,31,32]. Therefore, for devising a simple, at-home diagnostic test, a simple device, such as a smartphone, without any additional equipment, is required [33,34].

In a previous study, we developed QD-embedded silica NPs (SiO_2_@QD@SiO_2_; QD^2^) [35]. QD^2^ have a silica shell, which makes surface modification easy, and 200-fold stronger fluorescence intensity than a single QD. QD^2^ that exhibit high detection sensitivity have been used to develop a highly sensitive LFIA system [36]. In our previous studies, the LFIA system, which uses QD^2^, detected the HFF exosome 11-fold more sensitively than the conventional method, and thus, we deduced that QD^2^s were sensitive nanoprobes that could be used for LFIA. Additionally, QD^2^ with very bright fluorescence could ignore the fluorescence noise from the test strip. To date, QD^2^-based LFIA has neither been used for PSA detection nor combined with a smartphone. In this study, we developed a complete LFIA kit for detecting PSA with QD^2^. After the sample was loaded and developed onto an assembled LFIA strip, fluorescence intensity analysis was performed on an image acquired by a smartphone (iPhone 12, Apple Inc., Cupertino, CA, USA). Selectivity and stability tests were conducted to evaluate the new LFIA system. The proposed PSA LFIA kit is expected to be used by potential prostate cancer patients to analyze their PSA levels, without requiring complex analytical tools, and screen for the disease. 

## 2. Materials and Methods

### 2.1. Materials

Tetraethyl orthosilicate (TEOS), (3-mercaptopropyl)trimethoxysilane (MPTS), (3-aminopropyl)triethoxysilane (APTS), succinic anhydride, *N*,*N*-diisopropylethylamine (DIEA), 1-ethyl-3-(3-dimethylaminopropyl) carbodiimide hydrochloride (EDC hydrochloride), *N*-hydroxysulfosuccinimide (sulfo-NHS), 2-(*N*-morpholino)ethanesulfonic hydrate (MES), ethanolamine, Tween^®^ 20, polyvinylpyrrolidone (PVP, MW ≈ 10,000 Da), bovine serum albumin (BSA), sucrose, and alpha-fetoprotein (AFP) were purchased from Sigma Aldrich (St. Louis, MO, USA). Absolute ethanol (EtOH; 99.9%), aqueous ammonium hydroxide (NH_4_OH), and N-methyl-2-pyrrolidone (NMP) were purchased from Daejung (Sihung, Korea). Dichloromethane (DCM) was purchased from Samchun (Pyeongtaek, Korea). Polyethylene glycol (PEG, MW ≈ 400 Da) was purchased from Alfa Aesar (Haverhill, MA, USA). Amino polyethylene glycol acid (NH_2_-PEG-COOH, MW ≈ 600 Da) was purchased from Nanocs (New York, NY, USA). Anti-PSA (14801) antibody (Ab), anti-PSA (14803) Ab, goat anti-mouse IgG Ab, backing card, nitrocellulose (NC) membrane, absorbent pad, and cassette were purchased from Bore Da Biotech Co. Ltd. (Seongnam, Korea). Phosphate-buffered saline (PBS; pH 7.4) and tris-buffered saline (pH 8.0) were purchased from DYNE BIO (Seongnam, Korea). Deionized water (DW) was produced by using a Millipore water purification system of Vivagen (Seongnam, Korea). CdSe@ZnS QDs were purchased from Zeus (Osan, Korea). PSA for the standard experiment was purchased from Fitzgerald (Acton, MA, USA).

### 2.2. Synthesis of QD^2^

QD^2^s were synthesized by using a modified method presented in our previous study [35]. First, SiO_2_ NPs, which serve as the template of QD^2^, were prepared by using a modified Stöber method [37]. Briefly, 40 mL of EtOH, 1.6 mL of TEOS, and 3 mL of NH_4_OH were poured into a 100 mL round-bottom flask and stirred for 20 h at 25 °C. Then, SiO_2_ NPs were obtained after washing the solution several times with EtOH via centrifugation (8885 RCF, 15 min). Next, 8 mL of SiO_2_ NPs in EtOH solution (25 mg/mL, 200 mg), 200 μL of MPTS, and 40 μL of NH_4_OH were mixed and stirred vigorously for 12 h at 25 °C, to introduce the thiol group onto the surface of SiO_2_ NPs. Thiol-functionalized SiO_2_ NPs (SiO_2_-SH) were obtained after washing the solution several times with EtOH via centrifugation (8885 RCF, 15 min). To introduce CdSe@ZnS QDs onto the surface of SiO_2_-SH NPs, 4 mL of DCM, 800 μL of EtOH, 50 μL of DW, and 70 μL of QDs in toluene (100 mg/mL, 7 mg) were mixed in a 15 mL centrifugal tube and incubated in a shaking incubator for 3 h at 25 °C. Then, 50 μL of MPTS and 50 μL of NH_4_OH were added to the mixture. Next, the mixture was incubated in a shaking incubator for 3 h at 25 °C. To obtain QD-introduced SiO_2_ NPs (SiO_2_@QDs), SiO_2_@QDs were washed several times with EtOH via centrifugation (8885 RCF, 15 min). These SiO_2_@QDs were dispersed in 5 mL EtOH to adjust their concentration to 2 mg/mL (based on the initial concentration of SiO_2_ NPs). To coat the SiO_2_@QDs with silica, 50 μL of TEOS and 50 μL of NH_4_OH were added to 5 mL of SiO_2_@QDs in the EtOH solution (2 mg/mL, 10 mg). The mixture was incubated in a shaking incubator for 20 h at 25 °C. Finally, silica-coated SiO_2_@QDs (SiO_2_@QD@SiO_2_, QD^2^) were obtained after washing the mixture several times with EtOH and dispersing in 5 mL of EtOH (2 mg/mL).

### 2.3. Conjugation of Anti-PSA Abs onto QD^2^ (QD^2^-PSA Ab)

Anti-PSA Abs were conjugated onto QD^2^, as described in our previous study [36]. To aminate QD^2^, 10 μL of APTS and 10 μL of NH_4_OH were added to 1 mL of QD^2^ in EtOH (1 mg/mL, 1 mg) and incubated for 1 h at 25 °C. Next, QD^2^s were washed twice with NMP. After aminated QD^2^s were dispersed in 500 μL of NMP, 1.75 mg of succinic anhydride and 3.05 μL of DIEA were added into this mixture to introduce the carboxyl group. This mixture was incubated for 1 h at 25 °C. QD^2^s were washed several times with DW and dispersed in 700 μL of DW. Then, EDC/sulfo-NHS coupling was conducted to make active groups. At first, 100 μL of 2% (*w*/*v*) EDC hydrochloride in DW, 100 μL of 2% (*w*/*v*) sulfo-NHS in DW, and 100 μL of 500 mM MES in DW were added to the QD^2^ mixture and incubated for 30 min at 25 °C. QD^2^s were washed once with 50 mM MES and dispersed in 1 mL of 50 mM MES. After 10 μL of NH_2_-PEG-COOH (1.6 mM) was added to this suspension, the mixture was incubated for 2 h at 25 °C. These PEGylated QD^2^ were washed once with 50 mM MES and dispersed in 1 mL of 50 mM MES. Then, 3.2 μL of ethanolamine was added to the suspension to passivate the remaining active groups. After 30 min of incubation, the PEGylated QD^2^s were washed several times with DW and dispersed in 700 μL of DW. EDC/sulfo-NHS coupling was conducted again to generate active groups. Next, 150 μg of anti-PSA Ab (14803) was added to the suspension and incubated for 2 h at 25 °C. Anti-PSA Ab-conjugated QD^2^s (QD^2^-PSA Ab) were washed once with 50 mM MES and dispersed in 1 mL of 50 mM MES. The active groups were then passivated again. Finally, QD^2^-PSA Ab samples were dispersed in 1 mL of 0.5% (*w*/*v*) BSA in PBS.

### 2.4. Characterization of QD^2^ and QD^2^-PSA Ab

Transmission electron microscopy (TEM) images were acquired using JEM-F200 and JEM-1010 (JEOL, Tokyo, Japan). The ultraviolet–visible (UV–Vis) absorbance was measured using a UV–Vis spectrophotometer (Mecasys OPTIZEN POP, Daejeon, Korea). The photoluminescence (PL) intensity of NPs was measured by using a model Cary Eclipse fluorescence spectrophotometer (Agilent Technologies, Santa Clara, CA, USA), at an excitation wavelength of 385 nm.

### 2.5. Preparation of Test Strips

The LFIA kit was prepared by assembling the cassette, backing card, absorbent pad, conjugate pad, and sample pad. First, an NC membrane was prepared by spraying 1 mL of anti-PSA Ab (14801) in PBS (1 mg/mL) on the test line and 1 mL of goat anti-mouse IgG antibody in PBS (1 mg/mL) on the control line. The conjugate pad solution was prepared by adding 50 mg of BSA, 70 mg of sucrose, 20 mg of PEG (MW ≈ 400 Da), 1 mg of Tween^®^ 20, and 0.1 mg of QD^2^-PSA Ab to 1 mL of PBS (pH 7.4). The conjugate pad was then prepared by spraying 5 mL of the conjugate pad solution. The sample pad solution was prepared by adding 5 mg of Tween^®^ 20 and 25 mg of PVP (MW ≈ 10,000 Da) to 5 mL of 20 mM tris-buffered saline (pH 8.0). The sample pad was prepared by spraying 1 mL of the sample pad solution. Next, the NC membrane, absorbent pad, conjugate pad, and sample pad were attached to the backing card. Lastly, the prepared test strips were cut at a width of 4 mm and assembled with cassettes.

### 2.6. Preparation of Clinical Samples of Human Serum

This study was approved by the Seoul National University Bundang Hospital under trial registration number IRB No. B 1711/432-302. Human serum samples were collected according to the approved trial protocol. Participants of clinical samples are all men in their 20s and 50s. Clinical samples were prepared after separating serum with plasma from blood of 47 people. Additionally, the PSA level of serum samples was measured by using the ELISA method.

### 2.7. Analysis of PSA and Human Serum Using the Prepared Test Strips

For detecting PSA, the concentration of PSA was adjusted from 0 to 100 ng/mL with 0.5% Tween^®^ 20-containing PBS (0.5% PBST), and 100 μL of each sample was analyzed with individual LFIA kits. When developing clinical samples, 60 μL of serum and 40 μL of 0.5% PBST were mixed to facilitate the development of the samples.

### 2.8. Photoluminescence Intensity Measurement of the Prepared Test Strips

The images of all test strips were acquired using a smartphone (iPhone 12, Apple Inc., Cupertino, CA, USA) camera. The photoluminescence intensity was measured using ImageJ ver. 1.53a (National Institutes of Health, Bethesda, MD, USA). The red, green, and blue channels of the original images were split using the software, and only the intensity of the red channel was measured because red fluorescence QDs were used.

## 3. Results and Discussion

### 3.1. Characterization of QD^2^ and QD^2^-PSA Ab

QD^2^ are nanostructures in which numerous QDs are attached using silica as a template, and the QDs are coated with a silica shell for easy surface modification. QD^2^s were selected as probes in our facile lateral flow immunoassay (LFIA) system to detect PSA. It was assembled as described in our previous study [36]. The schematic of this process is shown in Figure 1a. First, SiO_2_ NPs, which are templates of QD^2^ with a size of 142.18 ± 6.47 nm, were prepared using the Stöber method (Appendix A). Before conjugating QDs with SiO_2_ NPs, a thiol group was introduced by treating the surface of SiO_2_ NPs with MPTS. Then, QDs were introduced onto the surface of SiO_2_ NPs based on the affinity between the QD and thiol groups [35]. In addition, SiO_2_@QDs were coated with a silica shell to prevent aggregation and ease the surface modification with various functional groups. Finally, QD^2^s were synthesized via surface modification for use in the LFIA. Anti-PSA Ab was conjugated to QD^2^ after PEGylation. NH_2_-PEG-COOH was added to prevent aggregation between NPs [38]. The TEM image showed that the QDs were introduced onto the SiO_2_ surface, and no aggregation was observed (Figure 1b). The size of QD^2^ was measured to be 190.13 ± 6.11 nm. The intensity of UV–Vis absorbance of NPs increased when QDs were attached, whereas the intensity did not change significantly when anti-PSA Ab was attached to QD^2^ through surface modification (Figure 1c). The PL intensity of SiO_2_ NPs, QD^2^, QD^2^-PSA Ab, and QDs was also investigated (Figure 1d and Appendix A). All concentrations were unified to the same level (particles/mL). Comparison of the PL intensities of single QDs and QD^2^ with the same number of particles indicated the very bright fluorescence of QD^2^ under UV light (Appendix A). Comparison of the PL intensities at 620 nm showed that the PL intensity of QD^2^ was 208-fold higher than that of single QDs (Figure 1e). These very bright QD^2^ make sensitive detection possible despite the background noise of the test strip. When the antibody was attached to QD^2^, the PL intensity did not markedly decrease (84.3% of original QD^2^).

### 3.2. Detection of PSA Using the QD^2^-Based LFIA Test Strip

The components of the LFIA kit for PSA detection and the overall analysis process are described in Figure 2a. The test strip had four major components: the sample pad sprayed with the sample pad solution, the conjugate pad sprayed with the conjugate pad solution and QD^2^-PSA Ab mixture, NC membrane sprayed with antibody solutions, and absorbent pad that absorbed the unbound developed sample. Preparation of the LFIA kit for PSA detection was completed by assembling the test strip into a cassette. The concentration of PSA was adjusted with 0.5% Tween^®^ 20-containing PBS (0.5% PBST), and 100 μL of each sample was analyzed with individual LFIA kits. For analyzing human serum, 60 μL of serum and 40 μL of 0.5% PBST were mixed. The loaded sample containing PSA was developed completely along the test strip in 15 min, and the results were visible on the NC membrane. To analyze the results, a photograph of the developed LFIA kit was acquired with a smartphone (iPhone 12, Apple Inc., Cupertino, CA, USA), and the RGB channels of the photograph were separated using ImageJ software. Since QD^2^ was used as a probe of LFIA emitting red fluorescence, PL analysis was performed using only the red channel. Green and blue channel images are shown in Appendix A. To confirm the LOD of the prepared kit, samples with various concentrations of PSA (0, 0.1, 0.3, 1, 3, 10, 30, and 100 ng/mL) were prepared and analyzed using the LFIA kit (Figure 2b). The results confirmed that non-specific binding did not occur in the absence of PSA, and the test line showed strong fluorescence intensity when the concentration of PSA was 100 ng/mL. For the quantitative analysis of PSA, fluorescence intensities of the test line (T value) and control line (C value) were measured using ImageJ. Since the C value decreases as the T value increases in LFIAs, the ratio of T to C values (T/C) is used for an accurate quantitative analysis [7]. Experiments were repeated three times at all concentrations, to obtain the T/C value at each concentration, and a fitting curve for the results was constructed (Figure 2c and Appendix A). The LOD of the developed LFIA kit was calculated as 0.138 ng/mL (R^2^ = 0.9865). To our knowledge, this LOD is the most sensitive result obtained to date for PSA detection using any LFIA kit based on fluorescence. Individuals with PSA concentration <2.5 ng/mL were considered to belong to a safe zone, while those with PSA concentration >2.5 ng/mL should seek medical advice (medical checkup zone) because the latter concentration implies an increased likelihood of developing prostate cancer [7].

### 3.3. Detection of PSA in Human Serum Using the QD^2^-Based LFIA Test Strip

Experiments were conducted to identify the LFIA kit that detects PSA both in the prepared PSA solution and in the human serum. To this end, serum samples from 47 participants with PSA concentrations of 0.001–12.950 ng/mL were analyzed using the developed LFIA system (Table 1, Figure 3a). For accurate analysis, the T/C value of each serum sample was calculated, and these are denoted as dots in Figure 3b. Most T/C values were less than 0.4 at PSA concentration <2.5 ng/mL (the criterion distinguishing the safe zone from the medical checkup zone) and higher than 0.4 at PSA concentration >2.5 ng/mL. Although there were some cases where the T/C values were >0.4 at PSA concentration <2.5 ng/mL (false positive), the T/C values of all samples in the medical checkup zone were less than 0.4 (false negative) because of the impurity of the clinical samples, which include proteins other than PSA. Based on these results, potential prostate cancer patients who require medical checkups can be identified using the T/C value. Since the process of obtaining the T/C value only requires a smartphone and a software program, without additional equipment, patients undergoing examination can easily self-diagnose. Based on the threshold T/C value of 0.4, a receiver operating characteristic (ROC) curve was generated (Figure 3c) to illustrate the diagnostic ability of this binary classifier system [39,40]. AUC was also measured to present the overall summary of diagnostic accuracy. The resulting AUC was 0.852 (0.5 represents a coin flip (random) and 1.0 represents perfect diagnostic accuracy). In the developed system, there is no concern of missing the optimal treatment stage due to false-negative results, and it is possible to screen rapidly without obtaining the false-negative results. However, there are some limitations in that the false-positive cases should be reduced, and 0.5% PBST solution is required to develop serum. To overcome these limitations, our group plans to reduce non-specific binding and facilitate the development of serum by adding new blockers such as casein to the conjugate pad.

### 3.4. Selectivity Test of the Developed LFIA System

A selectivity test was conducted to test whether the developed LFIA kit selectively detected PSA. AFP, which is one of the biomarkers of liver cancer, newborn calf serum (NCS), and PSA (100 ng/mL) were developed on separate strips (Figure 4a). The results showed that during PSA detection, both the test and control lines showed fluorescence, while only the control line showed fluorescence during AFP and NCS detection. To determine the accuracy of the analysis, the T/C values of each strip were calculated (Figure 4b) as follows: 1.033 (PSA), 0.026 (AFP), and 0.014 (NCS). This result shows that the developed LFIA kit can detect PSA selectively.

### 3.5. Stability Test of the Developed LFIA System

Clinical sample no. 31 (PSA concentration = 4.557 ng/mL) was analyzed using an LFIA kit, and the strip that was used for this analysis was photographed every day for 10 d to examine its signal stability (Figure 5a). As shown in Figure 5a, fluorescence intensity was maintained even after 10 d because of a low photobleaching characteristic of QD. To determine the accuracy of the analysis, the T/C value of the strip was calculated and normalized (Figure 5b). The results showed that the T/C value was maintained even after 10 d, and the relative standard deviation (RSD) was 6.16%. Collectively, these findings suggest that the fluorescence intensity does not markedly decrease and is sufficiently maintained for a long time after the system is tested.

## 4. Conclusions

In the present study, we developed a highly sensitive LFIA kit for detecting PSA using QD2. This LFIA kit can be used only with a smartphone and software without additional equipment. The developed LFIA kit had a LOD of 0.138 ng/mL, indicating the high sensitivity of the kit. In addition, 47 human serum samples were analyzed using the LFIA kit, and the results showed that the kit could be applied to actual clinical samples. Serum samples that contained more than 2.5 ng/mL of PSA could be identified after calculating the T/C value using this system, and thus, potential patients who need additional medical checkups can be identified. Although there were some false-positive results with PSA concentration <2.5 ng/mL, no false-negative results were obtained. The AUC value was calculated as 0.852. In addition, only PSA was detected selectively using the developed LFIA kit, and AFP and NCS were not detected. Furthermore, the fluorescence intensity did not significantly decrease for 10 d after the examination of fluorescence detected on the strip. The LFIA PSA test strip for prostate cancer screening developed in this study provides results rapidly and is easy to use; thus, patients can easily self-diagnose the disease without using complicated equipment.

## Figures and Tables

**Figure 1 nanomaterials-12-00033-f001:**
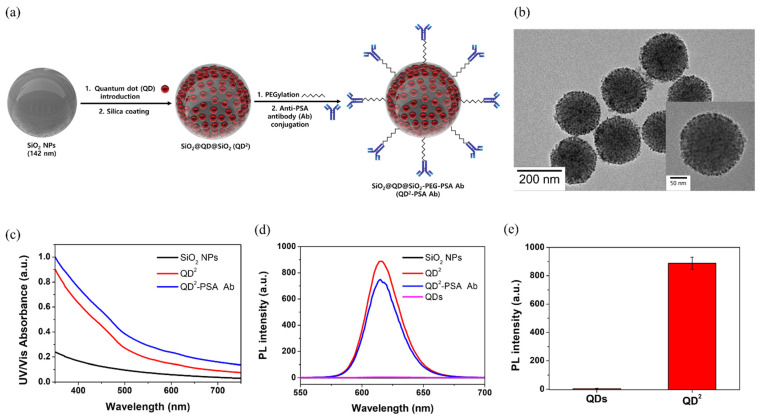
(**a**) Schematic illustration of QD^2^-PSA Ab fabrication (this figure is not drawn to scale); (**b**) transmission electron microscopy (TEM) images of QD^2^. The inset depicts individual QD^2^; (**c**) UV–Vis absorbance of SiO_2_ NPs, QD^2^, and QD^2^-PSA Ab; (**d**) PL intensity of SiO_2_ NPs, QD^2^, and QD^2^-PSA Ab; (**e**) PL intensity comparison between QDs and QD^2^.

**Figure 2 nanomaterials-12-00033-f002:**
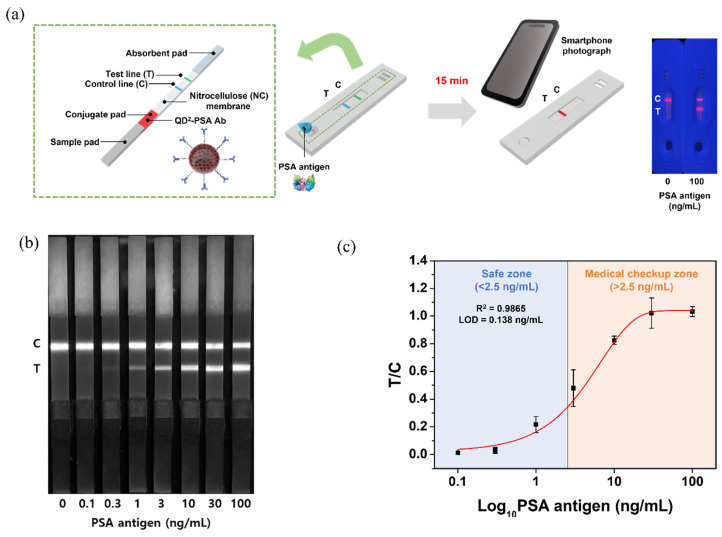
(**a**) Schematic illustration of lateral flow immunoassay (LFIA) process and precise analysis; (**b**) fluorescence image showing only the red channel of the test strip with PSA developed under a 365 nm UV lamp; (**c**) T/C value and fitting curve of the fluorescence intensity generated for each concentration of PSA.

**Figure 3 nanomaterials-12-00033-f003:**
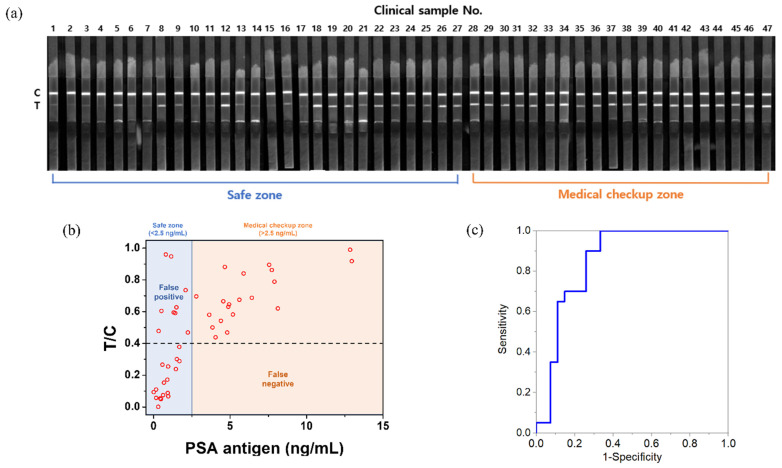
(**a**) Fluorescence image showing only the red channel of the test strip after the development of human serum samples under a 365 nm UV lamp; (**b**) T/C value and concentration of PSA of each human serum sample; (**c**) ROC curve.

**Figure 4 nanomaterials-12-00033-f004:**
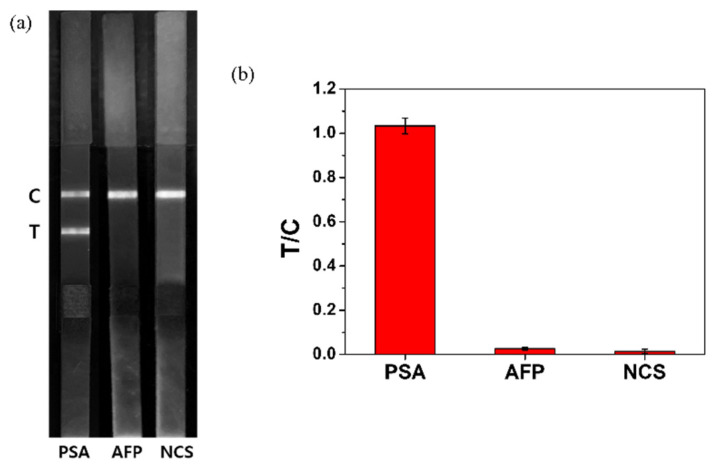
(**a**) Fluorescence image showing only the red channel of the test strip after the development of PSA, alpha-fetoprotein (AFP), and newborn calf serum (NCS) under a 365 nm UV lamp for the selectivity test; (**b**) T/C value of PSA, AFP, and NCS in strip tests.

**Figure 5 nanomaterials-12-00033-f005:**
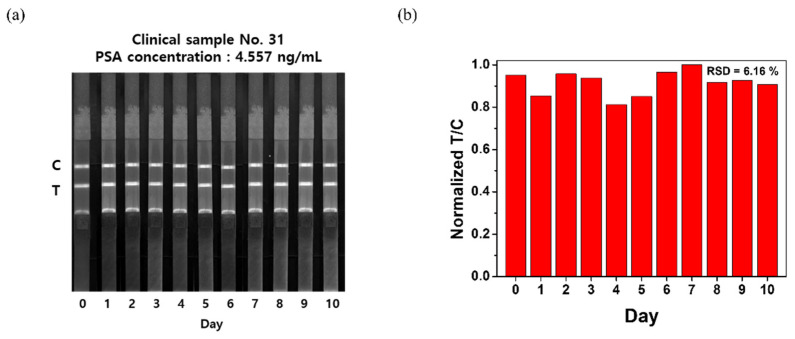
(**a**) Fluorescence image showing only the red channel of the test strip (stored for 10 d) after the development of 4.557 ng/mL of human serum sample under a 365 nm UV lamp for the stability test; (**b**) normalized T/C value of the developed human serum sample.

**Table 1 nanomaterials-12-00033-t001:** PSA concentration in clinical samples.

Clinical Sample No.	PSA Concentration (ng/mL)	Clinical Sample No.	PSA Concentration (ng/mL)	Clinical Sample No.	PSA Concentration (ng/mL)
1	0.001	17	0.954	33	4.557
2	0.159	18	1.146	34	4.655
3	0.164	19	1.309	35	4.815
4	0.300	20	1.412	36	4.888
5	0.323	21	1.455	37	4.931
6	0.429	22	1.488	38	5.182
7	0.479	23	1.514	39	5.607
8	0.514	24	1.677	40	5.880
9	0.577	25	1.689	41	6.418
10	0.619	26	2.093	42	7.551
11	0.677	27	2.233	43	7.729
12	0.797	28	2.788	44	7.900
13	0.897	29	3.637	45	8.125
14	0.919	30	3.847	46	12.843
15	0.921	31	4.043	47	12.950
16	0.945	32	4.398	-	-

## Data Availability

Data are contained within the article or Appendix A.

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
