# Peer review of "Lateral Flow Immunoassay with Quantum-Dot-Embedded Silica Nanoparticles for Prostate-Specific Antigen Detection"

_nanomaterials, 2021, doi:10.3390/nano12010033_

Round 1

Reviewer 1 Report

Sungje Bock et al developed a sensitive LFIA, using QD-embedded silica NPs for detection of PSA in prostate cancer serum, which require only a smartphone and computer software. They reported perfect diagnostic accuracy (AUC=0.916). In addition, the used LFIA only detects PSA and did not cross-react with AFP and newborn calf serum in the samples. Overall, this is an interesting attempt for potential point-of-care /rapid diagnostics of PSA detection for prostate cancer patients.

I have few comments and suggestions:

  1. It is not clear to me whether author included true clinical serum samples or 47 individual healthy serum spiked with PSA or 47 patients with known PSA values with commercial assay? What I understood that 47 healthy serum samples were spiked with different amount of PSA in PBS-T. Therefore, it is important to mention clearly these things in material section.
  2. If true clinical samples were used then what was Gleason score of prostate cancer patients and whether any among were benign (BPH) as well?
  3. What was the source of PSA used for standard in LFIA? Please mentioned the same in material section 2.1
  4. In conclusion, author intend to investigate there PSA level in urine via there LFIA but in my opinion determination of PSA in serum is more reliable than urine as not only benign but healthy person would also have huge amount PSA in urine along with prostate cancer patients. In my opinion detection of PSA glycoisoform could be useful rather than protein epitope/total PSA in urine as matrix.
  5. Did author tried the LFIA in whole blood and comparison of recovery with serum/plasma.
  6. I would recommend to discuss the limitation of study like sample size, volume of sample requirement at development stage, future planning to solve the false positive issue like use of blockers etc.

Reviewer 2 Report

  1. The authors say that the UV/Vis absorbance of SiO2 NPs, QD2 , and QD2 - 214 PSA Ab I the Figure 1c, but the wavelength is extended to the NIR range. Furthermore, the absorbance intensity is above the 1.0.
  2. Why is there no FL for the ODs?
  3. How do you calculate the LOD by the fitting curve?
  4. The results should be compared with the clinic standard methods and evaluated.

Round 2

Reviewer 2 Report

It would be better for the auhors to make corrections to minor methodological errors and text editing.
